# Assessing Mechanical and Rheological Properties of Potato Puree: Effect of Different Ingredient Combinations and Cooking Methods on the Feasibility of 3D Printing

**DOI:** 10.3390/foods9010021

**Published:** 2019-12-24

**Authors:** Iman Dankar, Amira Haddarah, Francesc Sepulcre, Montserrat Pujolà

**Affiliations:** 1Doctoral School of Science and Technology, Lebanese University, EDST, Hadath 1300, Lebanon; Iman.dankar565@gmail.com (I.D.); amirahaddara@hotmail.com (A.H.); 2Departament d’Enginyeria Agroalimentària i Biotecnologia, Universitat Politècnica de Catalunya, BarcelonaTech, 08860 Castelldefels, Spain; francesc.sepulcre@upc.edu

**Keywords:** microwaved, boiled, viscosity, thixotropy, microscopy, yield stress, 3D food printing

## Abstract

The effects of agar, alginate, butter, olive oil, and carrots on the mechanical and rheological properties of potato puree prepared by two different cooking methods (microwave heating (MP) and boiling (BP)) were investigated and interpreted in terms of starch microstructural changes. Microscopic observations revealed more aggregated and densely concentrated starch granules in MP samples. This consequently led to a significant increase (*p* < 0.05) in mechanical and rheological properties up to five times more than BP samples. All samples exhibited shear thinning non-Newtonian behavior. Butter proved its ability to maintain stiff network formation with starch molecules forming lipid-starch complexes characterized by high retention properties and increased stability due to high thixotropic and yield stress values. The pseudo-plasticity combined with high self-supporting ability (high yield stress and mechanical values) comprised by MP samples allows them to better behave during advanced technological processes such as extrusion 3D food printing.

## 1. Introduction

Potatoes are considered the third most consumed food crop world-wide, and today due to the shift towards more convenient nutritious food in ready-to-eat meal markets, vegetable potato purees would serve as a good alternative for money-rich time-poor countries, among which are the European countries. In fact, the unique functional properties of potato puree highlight it as a potential substrate in many advanced food technological processes, such as extrusion based 3D printing. Potato starch are characterized by large-sized granules comprised of versatile biopolymers; highly branched amylopectin and linear amylose chains, making them easily malleable to external stresses and to further modifications, whether mechanical (i.e., pressure, extrusion) or bio-chemical (i.e., incorporating additives, enzymes). Moreover, the abundant pectin substances in the middle lamella and the cellular wall of the potato cell facilitate the modulation of the textural properties of the starch under physical heat treatment (i.e., cooking) and widen its applicability as a thickening or gelling agent in the food technology sector [1]

Despite existing knowledge, accepting these perishable materials requires an intensive understanding of their physical properties. For example, studying the mechanical and rheological behavior is greatly necessary for developing an acquired product with the desired textural and quality characteristics. Additionally, this knowledge aids in predicting the analysis of process design and flow conditions in food processing and handling, such as in 3D printing (pump sizing, syringe size and length, total time of printing, extrusion, layer and conformation stabilization, etc.). In fact, 3D printing is an innovative additive manufacturing technology that aims to produce innovative shapes with customized food structures and personalized food nutrition sources for an individual person. All the newly published research in the field of 3D food printing focuses on optimizing and characterizing the properties of the printed substrate for achieving best printability [2,3]. For instance, Hamilton et al. [4] analyzes the rheological and structural properties of commercial breakfast spreads, Vegemite and Marmite, on the feasibility of 3D printing, while Severini et al. [5] linked the assessment of the rheological properties of cereal-based snacks enriched with edible insects to their compatibility with extrusion 3D printing.

Therefore, in food industrial applications, potato purees are generally mixed with different food substrates to enhance their functionality, stability, and texture as well as improve its performance during processing by modifying its rheological-mechanical properties to recommended values [6]. Determining optimum rheological properties is quite difficult since different substrates would interact differently with the complex potato puree system [7]. Therefore, in this study, various food substrates with different physico-chemical characteristics were incorporated. For example, fat substrates (butter and olive oil) were added to investigate the formation of starc-lipid complexes and their impact on the performance of the starch systems which are of great interest to food industries and for human nutrition [8]. Furthermore, butter and olive oil are considered an additional nutritional value for potato puree as a whole. Butter is rich in vitamins A, D, and E (i.e., antioxidants) and butyric acid, which has been used as a treatment for Crohn’s disease [9]. Olive oil is abundant in polyphenol antioxidants and vitamins E and K. it lowers bad cholesterol (i.e., LDL) in the blood and boosts immunity. Other highly consumed vegetables, such as carrots, rich in pectin and beta-carotene pigment, could be integrated to form a nutrient-denser potato puree, hence, allowing for the study of its physical properties as a new combined vegetable puree. Also, agar and alginate were incorporated, as hydrocolloids known previously to enhance the stability and texture of potato puree, mostly at a proportion of 1% [7]. Therefore, 1% of each of the abovementioned ingredients was incorporated in the sample preparation of potato purees in order for the results to be comparable. The aim of this work is to contribute to the knowledge that each substrate would impact on the mechanical and rheological properties of potato puree prepared previously, by two cooking methods (microwave and boiling), and to provide proper explanations for such effects based on microstructural-level interactions. Hence, this work aims to develop the functionality of potato puree and improve its textural properties for it to be incorporated into the innovative technology of extrusion 3D food printing.

## 2. Materials and Methods

### 2.1. Sample Preparation

Fresh potato tubers (*Solanum Tuberosum L cv Kennebec*) were purchased from a local supermarket. Based on the primary visual inspection, potato tubers composing almost the same size, shape, and rigidity were selected. Tubers were then washed, peeled, and cut into cubes of approximately equal sizes. Afterwards, 100 g of potato cubes were subjected to two different cooking treatments: Microwave heating and boiling. For microwaving, potatoes were placed in a Sivica Porcelain Square (specially for microwave cooking) and then cooked at 700 W for 6 min in the absence of water whereas for boiling, potatoes were set in a beaker filled with distilled water at a proportion of 2:1 (*w*/*w*) at 98 ± 2 °C for approximately 20 min. Olive oil, butter, alginate, and agar were then added separately at 1% *w*/*w* for each type-cooked potato. Carrot puree was incorporated at a proportion of 1/3 of the potatoes. Carrots were boiled or microwaved respective to whether they were added to boiling (BP) or microwave heating (MP) groups. For each cooking treatment, each formula was prepared 4 times with the mentioned ingredients. All samples were set in an incubator (for approximately 25 min) until they maintain a temperature of 20 °C preceding any measurements.

### 2.2. Microscopic Observations

To compare the structure and the alignment of the starch particles between raw, microwaved, and boiled potatoes, a thin film of each potato kind was spread on a glass slide and stained with diluted Lugo’s Solution. Subsequently, the preparation was examined under a compound light microscope (Nikon Trinocular Microscope Alphaphot2 s2, Tokyo, Japan). The micrographs were taken at 10× magnification.

### 2.3. Mechanical Characteristics

The mechanical characteristics (i.e., firmness (kg), consistency (kg s), and cohesiveness (kg)) of the different blends were tested using a textural analyzer (TA.XT Plus, Stable Microsystems, Godalming, UK) coupled with a back extrusion cell and a 35 mm disc. Samples were placed in a standard-sized cylinder. During the test, the disc penetrated a distance of 30 mm at a speed of 2 mm·s^−1^, (recording a maximum force) after which the probe returned to the original position [2]. The ‘peak’ or maximum force is taken as a measurement of firmness—the higher the value, the firmer the sample. The area of the curve up to this point reflects the measurement of consistency—the higher the value the, thicker the consistency of the sample. The negative region of the graph, produced on probe return, is as a result of the weight of sample which is lifted primarily on the upper surface of the disc on return due to back extrusion. The maximum negative force is taken as an indication of the cohesiveness of the sample—the more negative the value, the more ’sticky’ or’cohesive’ the sample [10]. Each sample was replicated at least 3 times.

### 2.4. Steady Rheological Measurements, Thixotropy, Yield Stress

The rheological measurements were performed in a viscometer (HAAKE Rheostress, Barcelona, Spain) controlled with commercial computer software (RheoWin3 Job Manager, Karlsruhe, Germany. Samples were analyzed for their flow properties using a concentric rotating cylinder (SV2). Steady rotational tests were performed to study the flow behavior, thixotropy, and yield stress. Yield stress was estimated as the point in which viscosity as a function of the shear stress (η = f(τ)) changed abruptly [11]. The temperature of the rheological tests was set constant at 20 ± 0.1 °C. The results were reported as the average of three replicates.

### 2.5. 3D Food Printing Conditions

A RepRap BCN3D+ extruder printer (CIM Foundation, Barcelona, Spain) coupled with a syringe tool of 100 mL volume and 4 mm nozzle diameter was used to print 3D samples of potato purees. The code for the desired 3D object was transferred through an SD card from a CAD program (CURA 15.02.01). Speeds set in the CURA program were as follows: Travel speed = 100 mm·s^−1^, infill speed = 40 mm·s^−1^, printing speed = 40 mm·s^−1^, flow % = 100, and retraction speed = 40 mm·s^−1^


Quality of 3D food printing was evaluated in terms of smooth printing (continuous extrusion without detachment), self-supporting structure (able to hold up its 3D shape properly post printing), and creamy surface appearance at the end.

### 2.6. Statistical Analysis

Statistical analyses of the data were conducted using Minitab 18 (Minitab Inc., State College, PA, USA). The data concerning textural characteristics were tested for significant differences (*p* < 0.05) using analysis of variance, one-way ANOVA, and Tukey’s HSD comparison test.

## 3. Results and Discussion

### 3.1. Microscopic Observations

Observations showed separate clusters of small oval starch granules scattered around the polygonal parenchyma cells in raw potato (Figure 1c) compared to well-defined and extensively swollen starch granules that occupied nearly the entire parenchyma cell volume in MP and BP (Figure 1a,b). Ormerod et al. [12] found that upon cooking, the relative size of potato starch cells increases compared to raw potato (RP) due to starch gelatinization when exposed to about 95 °C in excess water. Denser pigmented aggregates with a higher intercellular cohesion were observed among MP starch granules, which could be attributed to differences in the rate of heating or the method used for cooking [13].

Anderson et al. [14] reported that during boiling, soluble amylose starch leached out of the cells, reducing and destroying pectin substances found in the middle lamella and the cellular wall of the potato tissue, and ensuring a high uptake of water from the liquor to the inside tuber. This finding explains the reason behind the more separated starch granules and the less intense color observed in BP. The percentage of the relative concentration of starch per unit area of the cell lessened.

In addition, the water content present in starch governs the internal microstructure and affects its textural and rheological properties [15]. The lower water content of MP (704.8 g·kg^−1^) versus BP (822.8 g·kg^−1^) facilitated stronger gel formation, as reflected through the higher compacted microwaved starch granules observed under the microscope [3]. This difference in water content could be explained by the fact that during boiling, water from the surrounding medium can penetrate the potato tuber and increase the relative percentage of moisture content, in contrast to the microwaved cooked potatoes that were carried out without water, eliminating the possibility of water penetration into the potato media. Adding to this, the higher energy input that is applied during microwaving aids in a faster rate of water evaporation from the potatoes, resulting in relatively lower levels of moisture content.

### 3.2. Effect of Added Substrate on Mechanical Properties in Boiled and Microwaved Potatoes

The firmness, cohesiveness, and consistency of MP were significantly different (*p* < 0.05) from BP samples (Table 1), with MP alone possessing much higher values compared to BP, attributed to the lower moisture content expressed by the MP samples, which consequently provides MP with a firmer and more rigid like structure. These results are in agreement with [16], who reported that potato textural properties varied according to the cooking treatment, where microwaved samples provided firmer structures than boiled samples. Likewise, Singh et al. [15] linked changes in the textural characteristics of cultivars with their respective cellular arrangement (i.e., a packed cell alignment was much harder and cohesive than a loose cell arrangement). This finding is in accordance with the microscopic observations and explains why MP exhibited much higher mechanical characteristic values.

After the addition of the different additives, the range of the mechanical properties for all the MP samples persisted to be higher than that of the BP samples. However, some deviations in the mechanical properties were recorded according to the type of ingredient added. For instance, in MP, agar showed its ability to produce the highest increase in textural properties due to its effect of creating a strong network between the glycan chains, enhancing the particle–particle surface contact [7]. Alginate and butter differed significantly from agar, but no significant difference was detected between them in MP samples (Table 1). However, the fat substances added (i.e., butter and olive oil) exerted different effects on the mechanical properties of potatoes regardless of the cooking method used. Butter has a more compact structure where all its chemical bonds are saturated, yielding a higher ability to form hydrogen bond interactions with the starch molecules. Butter, thus, increased the mechanical properties of potatoes compared to olive oil, which is chemically unsaturated, containing fewer sites for interactions and, hence, inducing a more disruptive effect [17,18]. The addition of carrot or olive oil in the BP samples did not improve the mechanical properties.

### 3.3. Effect of the Type of Added Substrate and Cooking Treatment on Rheological Characteristics

#### 3.3.1. Viscosity

Rheological starch properties with different food additives and after application to different cooking treatments were studied through the behavior of viscosity curves. Flow curves (Figure 2) for all the potato samples exhibited an exponential decay for the shear viscosity versus shear rate, indicating a non-Newtonian, strong shear thinning behavior, in agreement with several authors [7,8,9,10,11,12,13,14,15,16,17,18,19]. Comparing Figure 2a,b, the first observation to be made is the huge difference between the viscosity axis ranges for the MP samples (500–3500 Pa·s) versus that of BP (100–900 Pa·s). Moreover, MP alone recorded an initial viscosity of ~1250 Pa·s, which is almost 10 times higher than that recoded for BP alone (~170 Pa·s). This outcome re-demonstrates the advanced internal structure and stability of all MP samples compared to that of BP. In this sense, Andersson et al. [14] related cohesiveness to the viscosity such that the lower the intercellular distance, the more cohesive the product and the higher the viscosity, in agreement with the microscopic observations (Figure 1).

Moreover, viscosity curves showed that the addition of olive oil substrate expressed the highest decrease in viscosity values for both MP and BP samples, compared to the other incorporated fat substance (butter) which was able to elevate the viscosity approximately two times more than that of BP and MP. This could be attributed to the chain length as well as the degree of unsaturation of fatty acids that had different effects on the formation of starch lipid complexes. These starch lipid complexes are favored more by the presence of lower degree of unsaturation and shorter chain length of fatty acids [8], which complies more with the structural characteristics of butter.

Additionally, only agar combined with MP showed a distinct viscosity shape (an initial increase in viscosity until reaching a peak, beyond which an abrupt decrease occurs) compared to the other samples (instant decrease in viscosity) (Figure 2a). This could be explained by the fact that agar with MP had formed a forte gelling network that is durable to breakdown. Therefore, upon applying the initial shear rate, the viscosity tends to increase due to its permanence, with more particle–particle contact interaction; until it reaches a maximum strength above which it cannot hold up the applied shear rate anymore; this peak corresponds to the viscosity value of agar with MP, ~3300 Pa·s.

#### 3.3.2. Yield Stress

Figure 3 shows the viscosity vs. shear stress for potato samples representing their yield stress. The yield stress is the critical stress level where viscosity rapidly decreased for all the samples, characterized by a change in microstructure since starch granules are unable to absorb more energy without being deformed [11]. The yield stress values for all of the samples are shown in Table 2.

In fact, BP samples expressed much lower yield stress values which could be attributed to the further scattered starch granules, acquiring a reduced fraction of force for cellular separation (Figure 3). In relation to the effect of additives’ incorporation on the yield stress of MP and BP, it was revealed that agar and butter possessed the highest yield stress values. This lengthening in stress yield could be explained by the fact that both agar and butter affected the starch’s internal microstructure deeply. In a previous study, Dankar et al. [7], reported that potato puree with 1% agar exerted a high yield stress of ~1000 Pa, similar in value to that exerted by MP + 1% agar (~1200 Pa), where agar had the ability of bridging between starch granules through hydrogen bonding, promoting their association and providing starch with better texture and appearance. On the other hand, butter insertion inside potatoes might induce two forms of interactions; first, a continuous starch–lipid complex network that could be formed by Vander Waals hydrophilic interactions and hydrogen bonding between the carboxylic group at the end of the fatty acid chains and that of starch molecules. Second, the interactions might be compromised by a tendency of self-association among fatty acids, due to the recrystallization ability of butter that enhances the solidification effect, creating a highly consistent and stable product [20,21].

Conversely, olive oil reduced the yield stress for both BP and MP. Olive oil has a malleable structure that facilitates its penetration inside the starch molecule, weakening the extent of bonding and producing a labile starch structure. In contrast, carrot addition maintained a similar yield stress regardless of the cooking treatment used. Nevertheless, the use of alginate caused a slight decrease in yield stress for MP while increasing that of the BP samples. This could be attributed to the compacted structure of MP granules stimulating more of the repulsive effect between the phosphorous groups on the potato and the anionic charges on the alginate, creating a less continuous network characterized by lower stability. Likewise, Liu et al. [3] reported that the addition of anionic gums, such as Xanthan, to potato decreased its internal strength due to the repelling forces between the negatively charged gum and the anionic chain structures of potato starch.

#### 3.3.3. Thixotropy

All potato samples exhibited a hysteresis loop but with varying areas, indicating that all samples possessed thixotropic behavior. Again, MP samples possessed higher thixotropic areas compared to BP (Table 2). It is assumed that the bigger the hysteresis loop area, the more energy is required to destroy the internal structure of the material responsible for the flow time dependence [22], which implies a stronger internal stability of the material itself.

Agar showed an ability to produce the highest thixotropic areas for both BP and MP. The major effect between both cooking treatments was detected when butter was added. Generally, butter exhibits a strong thixotropic nature due to the action of two types of bonds that govern the fat crystal network and, thus, the lipid starch complexes: Irreversible bonds contributing to network stiffness and reversible bonds via Vander Waals interactions that would be disrupted under the shear rate effect but recover slowly via recrystallization [21]. Moreover, when added to a more compact structure (i.e., MP samples), a higher surface contact was established between fat crystals and starch molecules, generating a higher hysteresis loop in MP + 1% butter versus BP + 1% butter.

MP + olive oil showed a higher hysteresis loop than MP alone, although olive oil demonstrated its ability to decrease the viscosity and yield stress of MP (Table 2). This could be explained by the fact that the hysteresis loop area is also dependent on the energy that is required to restore the material to its initial form. Since olive oil has the ability to penetrate and cause drastic deformations within the intermolecular starch structure, a higher energy would be required to reform the original state.

### 3.4. Effect of Different Ingredient Combinations on the Feasibility of 3D Printing Trials

In 3D food printing trials of the different puree combinations, it was shown that MP samples exhibited better printability. The relative low viscosity that MP samples hold at high shear rates (i.e., pseu-doplastic) in combination with the high cohesiveness and consistency values they possess facilitates their smooth continuous extrusion from a nozzle tip. Moreover, the stronger thixotropic and yield stress values expressed by MP samples enhances its internal material properties to reform rapidly once deposited and to minimize shape deformation under the hydrostatic pressure of consecutive layers [3,4,5,6,7,8,9,10,11,12,13,14,15,16,17,18,19,20,21,22,23]. Yet, some differences in the aspects of printing were spotted among MP samples (Table 3).

The incorporation of 1/3 carrot, 1% alginate, and 1% butter to the MP produced samples with similar viscosity ranges that provided smooth extrusion but printed end products with different degrees of stability, mainly related to differences at the level of yield stress and mechanical characteristic values. The lower mechanical characteristic values recorded by MP + 1/3 carrot made it more susceptible to deformation upon removal post-printing compared to alginate and butter samples, whereas the high yield stress value displayed by butter enhanced the self-supporting nature of the printed material while providing a creamy surface texture and attractive appearance (Figure 4). Similar results were observed by Lille et al. [24], who analyzed the effect of fat on printing performance by comparing skimmed milk powder, whose printability was sticky and hard, to semi-skimmed milk powder, which resulted in smooth printing with a precise product.

Conversely, BP samples generally exhibited softer pastes (lower mechanical and rheological characteristic values), resulting in an overflow of sample deposition and hampered printing details. All printed BP samples showed a compressed bottom layer due to their poor ability to resist the gravity of subsequent extruded layers. The effect is an increase in diameter and decrease of height of the targeted printed product (Figure 4d) compared to the sample MP + 1/3 carrot, (Figure 4c). This printing behavior among BP samples was observed to a less extent (printing got better) when 1% agar was added.

## 4. Conclusions

Microwaved and boiled potato samples possessed swollen gelatinized starch granules compared to raw potatoes, with microwaved potatoes revealing more aggregated and densely concentrated starch granules. The lower moisture content in microwaved potatoes provided a stronger inter-cohesive starch network, leading to higher values of mechanical and rheological properties.

Agar and butter substrates were able to elevate the mechanical properties (firmness, cohesiveness, consistency) as well as the viscosity, yield stress, and thixotropy of both MP and BP samples, due to the formation of a complex network between starch and additives.

The ability of butter to possess relatively low viscosity (pseudo-plastic behavior) and to re-solidify when left to rest at room temperature due to its high thixotropic values as well as its strong internal elasticity (possess high yield stress), widens the applicability of butter as a potential substrate to be used in advanced food technological processes such as extrusion-based 3D food printing.

BP samples showed smooth extrusion but with an overall deposition of material, characterized also by low textural properties to withstand the weight of the above printed layers, leading to hampered detailed-printed products.

MP samples showed better printability in terms of smooth continuous extrusion and product shape self-retention. Yet, the best printability was attained with MP + 1% butter, providing an elegant printed product with a smooth creamy surface.

The developed 3D printed products could be assessed as key formulas for delivering innovative shapes in restaurants or hotels and widen the applicability of more nutritive personalized colorful foods for children and elderly facing mastication-swallowing problems, thus promoting healthier snacks for them. 

## Figures and Tables

**Figure 1 foods-09-00021-f001:**
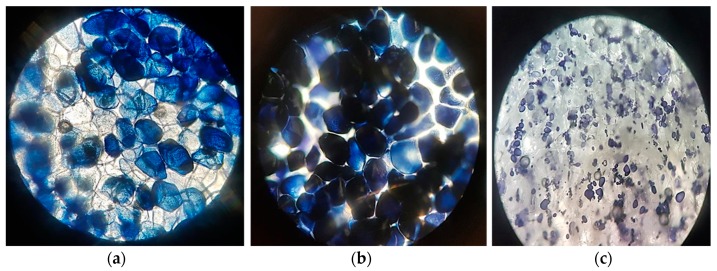
Microscopic images (10×) of (**a**) boiled potato, (**b**) microwaved potato, and (**c**) raw potato stained with Lugol’s iodine solution.

**Figure 2 foods-09-00021-f002:**
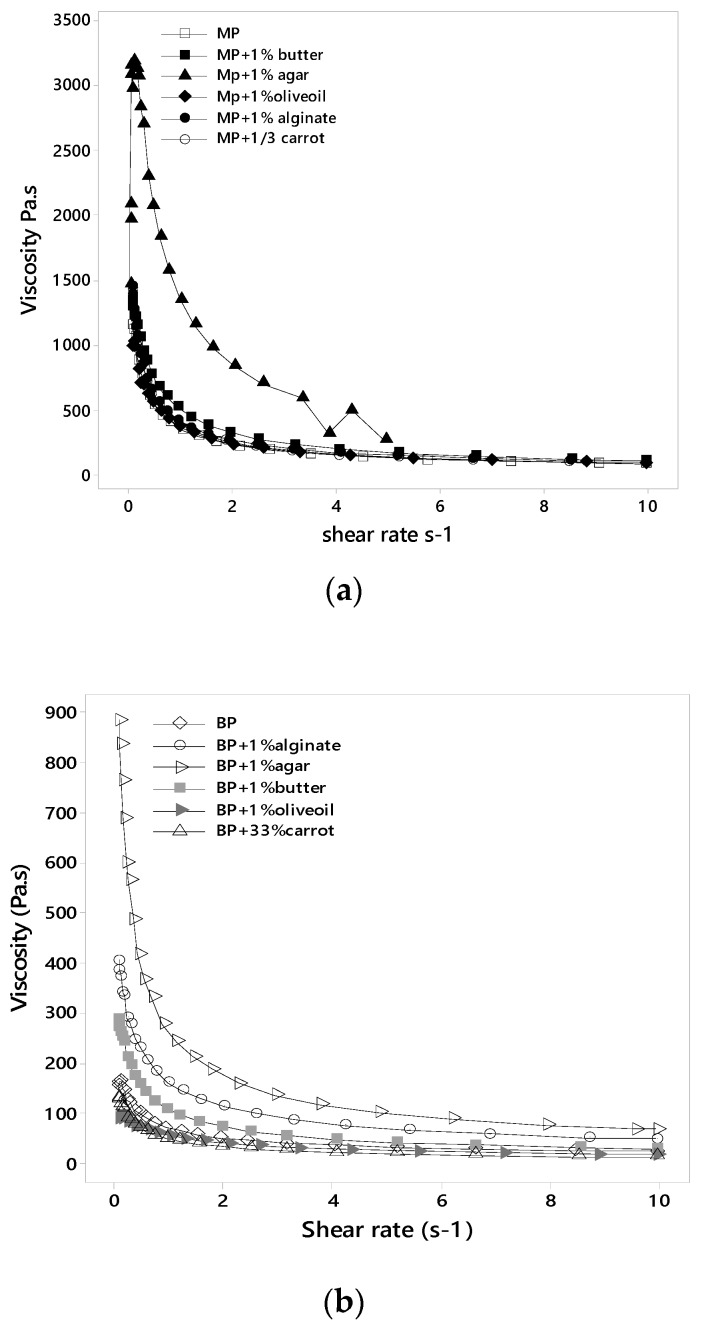
Typical flow curve of potato tubers with different substrates (**a**) microwaved and (**b**) boiled. Inset: flow curves at a shear rate below s^−1^.

**Figure 3 foods-09-00021-f003:**
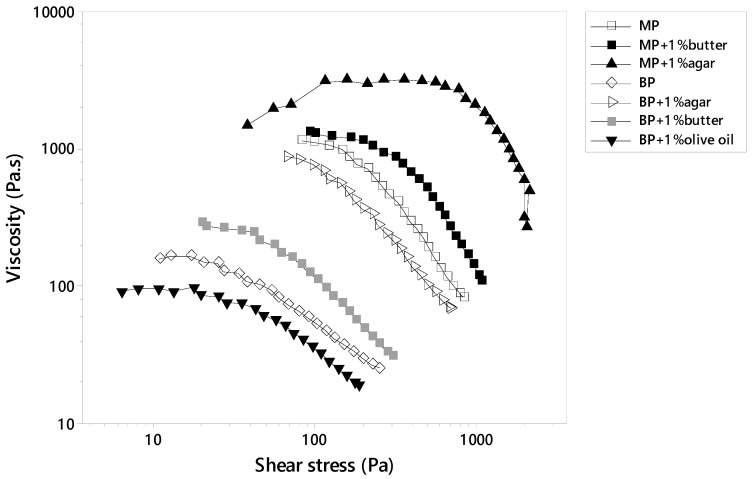
Measurement of the yield stress of microwaved (MP) and boiled (BP) potato with different substrates based on the stress ramp method.

**Figure 4 foods-09-00021-f004:**
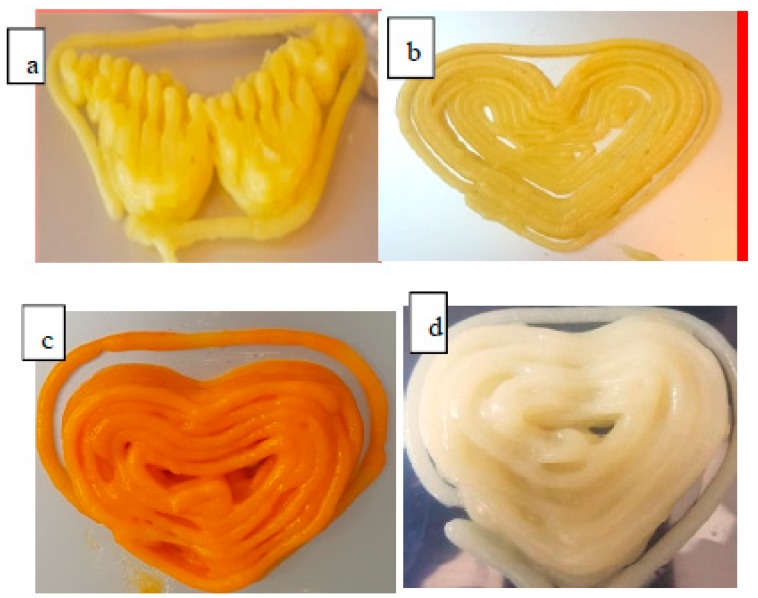
Printed products of microwaved potato with (**a**) 1% alginate, (**b**) 1% butter, (**c**) 1/3 carrots, and (**d**) boiled potato with 1% butter.

**Table 1 foods-09-00021-t001:** Mechanical characteristics values of firmness (Kg), consistency (Kg·s), and cohesiveness (Kg) of potato tubers with different types of food additives and after being exposed to two methods of cooking microwaving (MP) and boiling (BP).

Samples	Firmness (Kg)	Consistency (Kg s)	Cohesiveness (Kg)
MP	0.52 ± 0.03 ^d,e^	6.42 ± 0.65 ^d,e^	0.49 ± 0.004 ^d,e^
MP + 1% agar	1.54 ± 0.10 ^a^	18.40 ± 1.73 ^a^	1.55 ± 0.06 ^a^
MP + 1% alginate	0.80 ± 0.03 ^b^	9.87 ± 0.04 ^b^	0.73 ± 0.04 ^b^
MP + 1% butter	0.77 ± 0.02 ^b^	9.16 ± 0.30 ^b,c^	0.72 ± 0.01 ^b^
MP + 1/3 carrot	0.63 ± 0.04 ^c^	7.55 ± 0.11 ^c,d^	0.61 ± 0.05 ^c^
MP + 1% olive oil	0.58 ± 0.01 ^c,d^	7.09 ± 0.31 ^d,e^	0.55 ± 0.05 ^c,d^
BP	0.14 ± 0.004 ^g,h^	1.64 ± 0.09 ^f,g^	0.11 ± 0.006 ^g,h^
BP + 1% agar	0.43 ± 0.008 ^e^	5.51 ± 0.14 ^e^	0.42 ± 0.02 ^e^
BP + 1% alginate	0.27 ± 0.008 ^f^	3.35 ± 0.11 ^f^	0.24 ± 0.007 ^f^
BP + 1% butter	0.19 ± 0.003 ^f,g,h^	2.30 ± 0.06 ^f,g^	0.17 ± 0.006 ^f,g,h^
BP + 1/3 carrot	0.12 ± 0.006 ^h^	1.44 ± 0.07 ^g^	0.09 ± 0.005 ^h^
BP + 1% olive oil	0.11 ± 0.01 ^h^	1.36 ± 0.09 ^g^	0.09 ± 0.003 ^h^
Microwave carrot	0.51 ± 0.02 ^d,e^	6.08 ± 0.36 ^d,e^	0.47 ± 0.04 ^d,e^
Boiled carrot	0.23 ± 0.01 ^f,g^	2.70 ± 0.24 ^f,g^	0.19 ± 0.01 ^f,g^

Values are mean ± standard deviation (*n* = 3). Different letters (a–h) in the same column represents statistical differences (*p* < 0.05).

**Table 2 foods-09-00021-t002:** Values of thixotropy and yield stress of potato tubers with different types of food additives and after being applied to two methods of cooking microwaving (MP) and boiling (BP).

Microwaved Potato Samples	Thixotropy	Yield Stress (Pa)	Boiled Potato	Thixotropy (Pa·s^−1^)	Yield Stress (Pa)
	(Pa·s^−1^)		Samples		
MP	2231	280	BP	458.64	42
MP + 1% agar	8713	1250	BP + 1% agar	1791.67	200
MP + 1% alginate	2972	260	BP + 1% alginate	1105.78	120
MP + 1% butter	4633.5	330	BP + 1% butter	744.97	80
MP + 1/3 carrot	2866	280	BP + 1/3 carrot	371.73	40
MP + 1% olive oil	2885.25	245	BP + 1% olive oil	412.73	34

**Table 3 foods-09-00021-t003:** Effect of additives on microwaved potato samples feasibility during 3D printing.

Microwaved Potatosamples with	Advantages	Disadvantages
MP	Stable end product	Fill in of the shape not 100% ensured
Smooth extrusion
Withstand the printed shape over time
MP + 1% agar	Easily hand-able post printing	Poor fluidity
Precise definite dimension of layers	Retarded extrusion
Great resistance to compressed deformation	Non-continuous flow
Highest stable structured product for a long time post deposition	Rough surface structure
MP + 1% alginate	Stable end product with clearly observed details	Some plugging while extruding due to alginate coagulation ability
Layers coincide perfectly
Smooth surface
Withstand the printed shape over time
Hand-able post printing
MP + 1% butter	Smooth continuous extrusion	
Creamy surface
Proper arrangement of above layers
Retaining structure integrity
Withstand the printed shape over time
Removable

MP + 1/3 carrot	Soft surface	Works better with flat-base support product
Smooth continuous extrusion	Poor printing in fine-thin base supported product
Hold up the weight of the up deposited layers	Sticky
Stable product	Susceptibility to deformation upon removal
MP + 1% olive oil		Details are submissive
Less precise printing
Less stable end product
Spreads after printing
Non-hand able post printing

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
