# Peer review of "Assessing Mechanical and Rheological Properties of Potato Puree: Effect of Different Ingredient Combinations and Cooking Methods on the Feasibility of 3D Printing"

_foods, 2019, doi:10.3390/foods9010021_

Round 1

Reviewer 1 Report

General Comments:

Interesting paper but very descriptive.

English must be reviewed. Some sentences must be reworded.

In my opinion, the objective of the work must be clearly stated: to obtain a texture suitable for 3D-priniting.

Detailed comments:

Introduction:

The introduction paragraph is a little bit awkward and should be reworded. The objective of the reported work is not clear enough: texture improvement for 3D-printing, nutritional enhancement, microstructure analysis…. Line 25: Explicit why you consider potato puree as a high quality essential food. Not obvious, if you take into account nutritional aspects.

Materials and methods

Line 71: potato tubers were “selected”: how? Lines 74-77: Give more details for the ingredients Lines 87 and followings: storage duration between cooking and texture analysis? Which parameters on the curve gave you firmness, consistency and cohesiveness? Did you check the correlation of the rheological parameters with texture descriptors?

Results and discussion

Lines 132 and followings: the lower content of water in MP than in BP could be due to poor starch swelling during microwave cooking. Did you check swelling index? Did you check the gelatinization degree of starch granules? Discussion should be improved by taking into account scientific knowledge on starch , on the other ingredients used, and on interactions

Reviewer 2 Report

The most important notes:

1.) verse 32. The word 'enzymes' was used incorrectly. Under the influence of enzymes, starch undergoes biochemical rather than chemical modifications

2.) verses 32,33. Pectin substances in the middle lamella and the cellular wall are not found in starch (as it is shown in the context of your sentence), but in potatoes. This sentence should be corrected.

3.) verse 73. Microwave cooking is unclear. Were the samples placed in the microwave in the air or in water? If the potatoes in the microwave were cooked without water, this had a big impact on the mechanical properties. This should be included in the discussion.

4.) In the article, the authors use the words “cell” and “granules” interchangeably. The cell is found in potato, while the starch is in the form of granules. Potato cells contain starch granules, as it can be seen in the microscopic image of raw potato (fig 1); E.g. Verse 119 - should there be the word “cells”? In my opinion, this is about granules, not cells. Please check the entire text and standardize these terms.

Reviewer 3 Report

Dear authors,

I consider the manuscript "Assessing mechanical and rheological properties of potato puree: effect of different ingredients combinations and cooking methods" very innovative, which adds much to the concept of 3D printing from puree. However, I suggest some points:

1) Title: add the 3D printing aplication

2) Improve the introduction showing more works involving 3D printing, and the reason for to choice potato puree. Explain the importance of this work

3) Standardize references (Example line 55)

4) Add figures of the other purees in the Figure 4

Finally,  I recommend as "accept after minor revision".

Round 2

Reviewer 1 Report

remarks were taken into account, and modification were done